# Impact of Invasive Mechanical Ventilation on the Lung Microbiome

**DOI:** 10.3390/arm93040023

**Published:** 2025-07-01

**Authors:** Jose Luis Estela-Zape, Valeria Sanclemente-Cardoza, Maria Alejandra Espinosa-Cifuentes, Leidy Tatiana Ordoñez-Mora

**Affiliations:** 1Faculty of Health, Universidad Santiago de Cali, Cali 760035, Colombia; malejandraec@gmail.com (M.A.E.-C.); leidy.ordonez01@usc.edu.co (L.T.O.-M.); 2Health and Movement Research Group, Universidad Santiago de Cali, Cali 760035, Colombia; 3School of Public Health, Universidad del Valle, Cali 760043, Colombia; valeriasanclemente0@gmail.com

**Keywords:** humans, artificial respiration, mechanical ventilation, microbiota, dysbiosis

## Abstract

**Highlights:**

**What are the main findings?**

**What are the implications of the main findings?**

**Abstract:**

The lung microbiota is integral to maintaining microenvironmental homeostasis, influencing immune regulation, host defense against pathogens, and overall respiratory health. The dynamic interplay among the lung microbiota emphasizes their significance in shaping the respiratory milieu and potential impact on diverse pulmonary affections. This investigation aimed to identify the effects of invasive mechanical ventilation on the lung microbiome. *Materials and Methods*: A systematic review was conducted with registration number CRD42023461618, based on a search of PubMed, SCOPUS, and Web of Science databases, in line with the PRISMA guidelines. To achieve this, “(mechanical ventilation) AND (microbiota)” was used as the search term, replicable across all databases. The closing date of the search was 12 March 2025, and the evidence was scored using the MINORS scale. *Results*: A total of 16 studies were included, with patients aged 13.6 months to 76 years, predominantly male (64.2%). Common ICU admission diagnoses requiring invasive mechanical ventilation (IMV) included pneumonia, acute respiratory failure, and COVID-19. IMV was associated with reduced lung microbiota diversity and an increased prevalence of pathogenic bacteria, including *Prevotella*, *Streptococcus*, *Staphylococcus*, *Pseudomonas*, and *Acinetobacter*. The most frequently used antibiotics were cephalosporins, aminoglycosides, and penicillins. IMV-induced pulmonary dysbiosis correlated with higher infection risk and mortality, particularly in pneumonia and COVID-19 cases. Factors such as antimicrobial therapy, enteral nutrition, and systemic inflammation contributed to these alterations. *Conclusions*: Invasive mechanical ventilation has been associated with the development of alterations in the respiratory microbiome, resulting in reduced diversity of lung microorganisms.

## 1. Introduction

The pulmonary system is characterized by alveolocapillary junctions, which are essential for gas exchange between oxygen (O_2_) and carbon dioxide (CO_2_) under physiological conditions [1]. This system is divided into the upper and lower respiratory tracts and has a surface area of approximately 70 square meters, which is 40 times larger than the surface area of the skin [2].

In clinical microbiome research, the 16S rRNA gene is widely employed as the primary molecular marker for characterizing bacterial communities. Advances in microbiological analysis and gene sequencing technologies have yielded detailed insights into the respiratory tract microbiota, challenging the longstanding assumption that it is a sterile environment [3]. Microbial communities in the respiratory tract contribute to human health by colonizing and interacting with the respiratory epithelium, thereby influencing host immunity and barrier function [4]. These communities help defend against allergens and pathogens through the production of antimicrobial peptides and the regulation of epithelial cell proliferation [5]. Several studies [6,7] indicate that the lung microbiome in healthy individuals consists of diverse taxa, including Bacteroidetes, Proteobacteria, Firmicutes, and Actinobacteria. Frequently identified genera include *Streptococcus*, *Haemophilus*, *Prevotella*, *Veillonella*, and *Fusobacterium*.

Historically, respiratory microbiology has focused primarily on bacteria. External factors, such as environmental diversity, allergen/pathogen exposure, antibiotic use, and diet, significantly influence the composition of the host microbiota. Internal factors, including genetic predisposition, IgA-mediated immune response, mucus production, and innate/adaptive immune responses, also shape the microbiota [6,8]. A recent study [9] utilizing 16S rRNA sequencing on 2177 samples identified distinct microbiota profiles associated with acute respiratory distress syndrome (ARDS), hospital-acquired pneumonia (HAP), and mechanical ventilation. Alterations in the respiratory microbiome composition were observed, with a reduction in common microorganisms in HAP and mechanical ventilation, in contrast to an increase in pathogenic microorganisms in ARDS.

Invasive mechanical ventilation (IMV) is a critical intervention in intensive care units (ICUs) for managing respiratory failure, hypoxemia, and infections caused by microbial agents. However, while it provides essential respiratory support, IMV can significantly alter normal respiratory mechanics by affecting lung volumes, diaphragm function, and airway pressures, contributing to ventilator-induced lung injury (VILI) and impaired gas exchange [10]. Prolonged mechanical ventilation, particularly beyond 72 h, increases the risk of respiratory infections and promotes pulmonary dysbiosis, as early bacterial colonization of the oropharynx and endotracheal tube can occur within the first 12 h post-intubation [8,11]. Fromentin et al. (2021) [12] further suggest that the oropharyngeal and gastrointestinal microbiota serve as reservoirs for pathogenic bacteria, exacerbating microbial imbalance and inflammation during IMV (Figure 1).

Figure 1 presents a schematic representation of the impact of invasive mechanical ventilation on the lung microbiota. Phase 1 (blue) represents the healthy lung, characterized by a diverse and balanced microbial community. Phase 2 (yellow), corresponding to the initiation of mechanical ventilation, involves microbial translocation from the oropharynx to the lower airways and an early decline in microbial diversity. Phase 3 (red), associated with prolonged ventilation, is marked by established dysbiosis, characterized by a significant loss of diversity and a predominance of opportunistic pathogens. Colors are used to illustrate the progression of microbial imbalance and do not denote biological classifications.

The role of dysbiosis in chronic respiratory diseases or their exacerbation, especially in mechanically ventilated patients, remains unclear. Fromentin et al. (2021) [13] identified significant microbial populations in bronchial aspirates from mechanically ventilated patients, suggesting a potential role in the etiology of ventilator-associated pneumonia (VAP). Conversely, Emonet et al. (2019) [14] proposed that bacterial diversity decreases with prolonged invasive mechanical ventilation, leading to lung microbiome dysbiosis and the development of VAP. Zakharkina et al. (2017) [15] observed a variety of resident bacterial species in the lungs of intubated patients, originating from the oral cavity and migrating to the distal airways.

Despite substantial advancements in understanding the relationship between invasive mechanical ventilation and lung microbiota in critically ill patients, a significant gap in the current scientific evidence remains. This gap is characterized by the lack of a detailed analyses regarding the precise effects of mechanical ventilation on the composition and dynamics of the lung microbiota. This knowledge deficiency represents a major challenge, hindering the development and implementation of effective clinical strategies in ICUs. Therefore, there is a critical need for more in-depth research in this field to improve clinical outcomes and the management of critically ill patients undergoing invasive mechanical ventilation. Consequently, the objective of this systematic review is to elucidate the effects of invasive mechanical ventilation on lung microbiota.

## 2. Materials and Methods

This systematic review adheres to the guidelines outlined by the PRISMA (Preferred Reporting Items for Systematic Reviews and Meta-Analyses) statement. [16]. A Protocol record with the identifier CRD42023461618 was generated in PROSPERO.

### 2.1. Search Strategy

#### 2.1.1. Search Sources

The search was conducted in three databases: PubMed, SCOPUS, and Web of Science. It was completed on 12 March 2025 and included all eligible studies available up to that date.

#### 2.1.2. Research Question

The research question was formulated according to the PICO framework: “What are the effects of invasive mechanical ventilation on the pulmonary microbiota in humans with intra- or extrapulmonary pathologies?” This question was determined by the following components:-Population: Humans with intra- or extrapulmonary pathologies-Intervention: Invasive mechanical ventilation-Comparison: Not applicable-Outcome: Pulmonary microbiota

#### 2.1.3. Search Terms

To construct the search query, a standardized language was established in DeCS/MeSH terms and logical operators such as “OR” and “AND”, allowing for the creation of the search equation adjusted to the research problem: ((((humans) AND (artificial respiration)) OR (mechanical ventilation)) AND (microbiota)). The search query used in the three databases is summarized in Table 1.

#### 2.1.4. Eligibility Criteria

All primary research studies that described the effects of invasive mechanical ventilation on the lung microbiota in English, without limitations regarding the date, were included.

#### 2.1.5. Exclusion Criteria

Studies were excluded if they did not directly address the research question, involved animal models, were reviews, gray literature, theses, expert opinions, letters to the editor, or presented incomplete data. Articles with low methodological quality or those published in languages other than English were also excluded. The exclusion of gray literature was applied to ensure only the inclusion of studies with established peer-review standards and methodological rigor.

### 2.2. Study Selection

Data from the selected articles were independently extracted by two investigators (VSC, JLEZ) and subsequently verified by two additional investigators (JLEZ, LTOM). Initially, a review of the titles and abstracts of the studies was conducted to eliminate duplicates. Subsequently, a comprehensive review of the remaining articles was performed, and those that met the eligibility criteria were selected. In cases of discrepancies, the authors reached a consensus through discussions.

### 2.3. Quality Assessment

The MINORS scale [17] was employed to assess the methodological quality of studies that were not randomized controlled trials. This instrument comprises eight core items, with an additional four items applied when the study includes a comparison group, for a total of twelve. The scale evaluates methodological aspects such as prospective data collection, sample size calculation, and blinded outcome assessment. In studies with two groups, additional criteria include the inclusion of an appropriate control group, baseline equivalence between groups, and adequacy of statistical analyses. Two independent reviewers applied the MINORS scale in a blinded manner to each other’s assessments. Studies that did not meet the minimum methodological quality threshold were excluded from the review.

### 2.4. Data Extraction and Synthesis

The authors independently extracted data from the included studies, taking into consideration the summaries, methodology, results, and conclusions. Likewise, narrative methods were applied to synthesize the collected data. Flowcharts were used to organize the articles that addressed the research objective, and tables were used to describe the effects of invasive mechanical ventilation on respiratory tract microbiota.

## 3. Results

The initial database search identified 492 articles. After duplicates, 477 records were examined. Inclusion and exclusion criteria were applied to full papers, leading to the inclusion of 19 articles (Figure 2). However, 20% were excluded due to a methodological quality score of less than 10 on the MINORS scale [17]. Consequently, 16 articles were included in the study.

### 3.1. Characteristics of the Included Studies

The studies included in this review provided comprehensive data on the effects of mechanical ventilation on the lung microbiota, mortality, study populations, respiratory pathologies upon ICU admission, microorganisms detected, antibiotics administered, and their impact on microbiota.

The 16 studies collectively demonstrate that invasive mechanical ventilation is associated with a reduction in microbial diversity, an increase in the prevalence of pathogenic bacteria, and higher mortality rates [18,19,20,21,22,23,24,25,26,27,28,29,30,31,32,33]. Additionally, the antibiotics administered during mechanical ventilation can negatively affect lung microbiota. Antibiotics target both pathogenic and beneficial microorganisms, potentially leading to pulmonary dysbiosis.

Pulmonary dysbiosis induced by mechanical ventilation results in an imbalance of the lung microbiota, which can increase the risk of infections, inflammation, and pulmonary fibrosis.

### 3.2. Methodological Quality

The articles included in this review spanned between 2018 and 2025. Quality assessment using the MINORS scale revealed that 93.75% (15 studies) achieved scores between 16 and 24, indicating good study quality. However, one study (6.25%) scored 14, below the ideal threshold, primarily due to its non-comparative design. A key limitation across the studies was the absence of blinding (Figure 3).

An analysis of specific criteria revealed methodological weaknesses. Seven studies were rated as showing a high risk of bias for the “unbiased evaluation of endpoints” item, with an additional three studies being classified as unclear, indicating potential inconsistencies in outcome assessments. Additionally, four studies showed a high risk of bias regarding the “prospective calculation of sample size,” due to the absence of sample size estimation or inadequate reporting.

Blinding was not implemented in the majority of studies, representing a frequent limitation attributable to the nature of the study designs.

### 3.3. Demographics of the Included Studies

A total of 16 articles were included in this review, aligning with its objective. Table 2 summarizes key study characteristics, including methodology, study population, ICU-admitted pathologies, and mortality rates. This compilation facilitated a critical evaluation of the breadth and quality of the evidence.

The study population ranged in age from 13.6 months to 76 years. The majority of patients were male (64.2%), with females comprising 35.8% of the total cohort.

The study population ranged in age from 13.6 months to 76 years, with a higher prevalence of male patients (64.2%) compared to female patients (35.8%).

The primary reasons for ICU admission requiring invasive mechanical ventilation included lung diseases, lower respiratory tract infections, acute respiratory failure, and pneumonia, as reported in the studies [18,20,21,22,24,25,27,28]. A notable finding was the high mortality rate in patients with pneumonia who experienced ventilatory failure during extubation, reaching 63.2%, as observed in one of the studies [25].

### 3.4. Comparison with No Intervention

In fifteen studies, the involvement of a control or comparison group regarding lung microbiota was not documented. The absence of treatments allowed for a natural, unaltered observation of the respiratory microbiome. These studies, which did not implement specific interventions, focused on the presence of microorganisms in tracheal secretion culture samples, with an emphasis on the diversity of bacteria, viruses, and fungi in the respiratory tract. These findings provide a foundation for investigating the relationship between lung microbiota and mechanical ventilation, offering insights into how mechanical ventilation may influence the microbial composition and diversity of the lungs.

### 3.5. Prevalence of Bacterial Families in ICU Patients

A notable finding was the nearly ubiquitous presence of bacterial families with the potential to cause pneumonia in ICU-admitted patients. This result highlights the clinical importance of accounting for microbiological diversity in this critical environment and reinforces the necessity for management and treatment strategies that address the complexity of bacterial populations in these patients.

### 3.6. Dynamics of Pulmonary Microbiome Research

Research on the pulmonary microbiome primarily focuses on changes in the relative abundances of microbial communities. These variations reflect the dynamic balance of ecological forces within the lung. This key aspect emphasizes the complexity of the pulmonary microbiota and underscores the importance of considering not only the presence of microorganisms but also their relative abundances in the context of lung health and disease in critically ill patients.

### 3.7. Antibiotics Used

It was determined that 75% *(n* = 9) of the studies described the applied antimicrobial therapies for the treatment of various pathologies, including acute respiratory failure [21], pneumonia [22,23,29,30], head trauma [24], COVID-19 [25,28,33], sepsis [31], and neurological, respiratory, cardiac, or gastrointestinal conditions [27,32].

The most used class of antibiotics was cephalosporins, which included carbapenems, cefotaxime, ceftriaxone, cefoperazone + sulbactam, ceftazidime, and cefepime [22,24,27,28,29,30,31,32,33]. Notably, patients with head trauma who received enteral nutrition and empirical treatment with cephalosporins had a higher 28-day mortality rate (45%) [24].

Following cephalosporins, aminoglycosides such as amikacin, gentamicin, and tobramycin were the second most frequently used class of antibiotics [23,27,29], followed by penicillins, including oxacillin and piperacillin–tazobactam [25,28,29]. However, it was observed that 33.3% of patients with ventilator-associated pneumonia and carbapenem-resistant *Acinetobacter baumannii*, treated with amikacin, gentamicin, and piperacillin–tazobactam, died within 28 days [29].

The mortality rate of patients treated with beta-lactams for pulmonary pathologies was not reported [19], while the mortality rate for patients with aspiration pneumonia was 13.6% [22].

In patients with pneumonia [23,29] and COVID-19 [28,30], the most frequently prescribed antibiotics included fluoroquinolones such as ciprofloxacin, levofloxacin, linezolid, vancomycin, sulfamethoxazole, imipenem, and meropenem. In contrast, antimicrobial therapy with tetracycline, tigecycline, fosfomycin, and metronidazole was used exclusively in patients with *Pseudomonas aeruginosa* ventilator-associated pneumonia [23], while trimethoprim was administered to patients with ventilator-associated pneumonia.

Systemic antibiotics were exclusively administered to patients with acute respiratory failure [20]. Additionally, colistin and amphotericin B were used in patients with neurological, respiratory, cardiac, or gastrointestinal conditions [27].

### 3.8. Microorganisms Detected

A total of 28 microorganisms were detected in the entire cohort, with 23 being bacteria (82.1%) [18,19,20,21,22,23,24,25,26,27,28,29,30,31,32,33], followed by 4 viruses (14.2%) [19], and 1 fungus (3.6%) [27].

### 3.9. Bacterial Distribution According to Gram Staining

#### 3.9.1. Gram-Positive Bacteria

Gram-positive organisms accounted for a significant proportion of the isolates. *Streptococcus* spp. were the most prevalent within this group, representing 25% of identified microorganisms. These were associated with extrapulmonary conditions [19], acute respiratory failure [21], pneumonia [22,23,26], and COVID-19 [24,27].

*Staphylococcus* spp. accounted for 17.8% of the isolates and were identified in ICU patients with surgical pathologies, trauma [7,24], acute respiratory failure [21], and COVID-19 [25,28].

Less common gram-positive genera included *Helcococcus*, *Stenotrophomonas* [19], and *Olsenella* [25]. Across several studies, additional gram-positive species were collectively reported in 7.14% of cases [18,19,20,23,26,28].

#### 3.9.2. Gram-Negative Bacteria

Among gram-negative bacteria, *Prevotella* spp. were the most frequently identified, accounting for 28.5% of total isolates. These organisms were detected in the lower respiratory tract [19] and were linked to acute respiratory failure [21], pneumonia [23,26], and extrapulmonary diseases [20].

*Pseudomonas* spp. represented 14.2% of isolates, commonly identified in ICU patients with varied conditions [18], including acute respiratory failure [21] and pneumonia [23,26].

Both *Acinetobacter* spp. [23,24] and members of the *Bacteroidetes* phylum [23] each comprised 10.7% of the total microorganisms.

Other gram-negative species collectively accounted for 3.5% of the isolates. These included *Haemophilus influenzae*, *H. haemolyticus*, *Moraxella*, *Dolosigranulum pigrum*, *Veillonella* [19], and *Citrobacter* [23]. Additional gram-negative bacteria were reported in 7.14% of cases across various studies [22,23].

### 3.10. Viruses

Four main types of viruses were identified in children admitted to the pediatric ICU who required invasive mechanical ventilation due to lower respiratory tract infections, including those caused by rhinovirus, coronavirus, respiratory syncytial virus, and adenovirus [19].

### 3.11. Fungi

The results obtained through the testing of bronchoalveolar secretions in 53 patients with various neurological, respiratory, medical, cardiac, or gastrointestinal conditions who underwent mechanical ventilation showed that *Candida* spp. was the most frequently identified fungus, regardless of antimicrobial therapy or patient characteristics [27].

### 3.12. Impact on the Lung Microbiota

The health of the lung microbiota depends on the balance between the different bacteria present in the respiratory tract. The selected studies highlight the importance of maintaining a healthy lung microbiota, which acts as a protective system against pathogens [20,21,23,24,27] and can serve as an indicator of the severity of disease and mortality [17,18,21,26,28,31,32,33]. Additionally, analysis of lung microbiota has been used as a basis for the development of new approaches in the treatment of lung infections, particularly in cases of pneumonia [22,25,29,30].

Alterations in the lung microbiota are common in critically ill patients. Mechanical ventilation, antimicrobial use, gastric acid suppression, and enteral nutrition are events that can decrease this microbiota [18], leading to pulmonary dysbiosis. This change affects the interaction between the host and other bacterial cells, increasing the lung’s susceptibility to developing infections [24].

### 3.13. Effects of Mechanical Ventilation on the Lung Microbiota

Factors such as microaspiration of gastric and oral contents, altered body posture, oxygen toxicity in hypoxemic patients, impaired airway clearance, and ventilator-induced lung injury can reduce microbial diversity in the lungs, potentially contributing to an unfavorable prognosis [30,31]. The selected studies consistently report that mechanical ventilation influences lung microbiota. Invasive mechanical ventilation is a commonly employed therapeutic intervention in hospital settings to restore gas exchange. However, prolonged use, particularly beyond 72 h, can disrupt the respiratory microbiome by reducing microbial diversity in the lungs, leading to pulmonary dysbiosis [19,20,22,32].

Additionally, mechanical ventilation has been associated with systemic inflammatory responses and adverse clinical outcomes [21]. Thus, prolonged mechanical ventilation is shown to negatively impact lung microbiota composition [30,31,32].

These findings suggest that mechanical ventilation may significantly affect both the composition and function of the lung microbiota, potentially contributing to a range of clinical complications.

## 4. Discussion

This systematic review sought to examine the effects of invasive mechanical ventilation on the lung microbiota. The findings indicate that prolonged use of invasive ventilation leads to significant alterations in microbial diversity within the respiratory tract, which may increase susceptibility to pathogenic colonization, exacerbate pre-existing pulmonary conditions, and elevate mortality risk. Notably, the reviewed studies did not report consistent differences related to patients’ sex or initial clinical diagnosis, suggesting that these effects may be broadly applicable across ICU populations.

In all the included and analyzed studies, endotracheal aspirate samples were employed in mechanically ventilated patients, demonstrating a close association between the microbiota of the upper and lower respiratory tract. However, Qi et al. (2018) [23] postulated the potential translocation of lung microbiota to the intestine, with the ensuing effect of altering the intestinal microbiota and triggering several pathologies. Furthermore, the study by Kim SH et al. (2020) [5] provided evidence to support this association by demonstrating that the digestive and respiratory systems are intricately interconnected. This indicated that the presence of digestive fluids in the lungs has the potential to alter microbiota in both systems, which, in turn, could make the host vulnerable to multiple disorders owing to an impaired immune system.

Moreover, previous investigations [20,24,29] based on case–control studies agreed that critically ill patients present a significant decrease in the diversity of the respiratory tract microbiota compared to healthy individuals. A higher prevalence of genera such as gram-negative *Prevotella* (28.5%), *Streptococcus* (25%), and *Pseudomonas aeruginosa* (14.2%) has been found in the microbiota of patients with ventilator-associated pneumonia. These findings indicate a close correlation between the severity of lung infection and the composition of both gram-positive and gram-negative bacteria.

Other authors [25,28], however, have conducted research with patients who presented with COVID-19 and received invasive mechanical ventilation. In their investigations, they identified *Staphylococcus aureus* (17.8%) as the predominant pathogen in microbiota, which increases the risk of developing pneumonia and bloodstream infections. However, Shajiei et al. (2022) [27] demonstrated that the prolonged application of invasive mechanical ventilation promotes the colonization of fungi, mainly *Candida* spp., and that the presence of these fungi contributes to the development of ventilator-associated pneumonia.

Mortality rates have been shown to be higher among patients who develop ventilator-associated pneumonia. Qi et al. (2018) [23], in their study, showed a mortality rate of 29.6% among evaluated patients, particularly associated with infections related to Pseudomonas aeruginosa. Furthermore, De Pascale et al. (2021) [25] reported a mortality rate of 25.0%, with a higher incidence in patients admitted to the ICU. Conversely, previous studies [19,20,21] did not record any deaths. This analysis demonstrates that mortality is directly related to the cause of the patient’s intubation and the prolonged use of mechanical ventilation, with higher rates noted in patients who develop ventilator-associated pneumonia.

A notable finding arising from the Bray–Curtis dissimilarity analysis [18,19] was the marked differentiation in the composition of the microbiota between critically ill patients under mechanical ventilation and healthy patients. This phenomenon translates into greater bacterial diversity within the lower respiratory tract of critically ill patients, which is unfortunately associated with an increase in the probability of in-hospital mortality compared to those with a more uniform microbiota. These results support the conclusions of Man et al. (2019) [19], wherein an association between the microbiota of the upper respiratory tract and lower respiratory tract infections was identified. It was observed that gram-negative bacteria can promote proinflammatory responses of the mucosa, which accelerates the process of pathogen spread.

Data from the study conducted by Kitsios et al. (2020) [21] also demonstrated that the reduction in lung microbiota diversity associated with invasive mechanical ventilation contributes to the development of pulmonary dysbiosis. This finding is consistent with previous observations reported by Fromentin (2021) [12] and Xue-Meng (2023) [34], where such changes are described in detail, including the potential for microbiota regeneration and its implications through the gut–lung axis [35]. This phenomenon is associated with systemic inflammatory responses and adverse clinical conditions. Furthermore, Alagna et al. in 2023 [20] showed that lower diversity in the upper respiratory tract microbiota before invasive mechanical ventilation could contribute to the development of pulmonary dysbiosis or ventilation-related effects. However, it is essential to emphasize that the number of publications exploring the association between the duration of mechanical ventilation and the development of dysbiosis in other segments of the body is insufficient to establish this relationship with certainty.

It is important to highlight the presence of potential confounding factors that were not adequately controlled in several of the included studies. Although this review emphasizes invasive mechanical ventilation (IMV) as a factor associated with pulmonary dysbiosis, the concomitant use of commonly administered antibiotics in intensive care unit (ICU) patients represents a variable that may independently alter the microbiota. Furthermore, other relevant factors—such as underlying comorbidities—were not consistently accounted for or adjusted in the analyses, as they were not reported in the majority of the studies.

Examining the pulmonary microbiota in patients undergoing invasive mechanical ventilation is a critical step toward developing innovative approaches to treat pulmonary infections and mitigate pulmonary dysbiosis. Evidence suggests that the composition of the pulmonary microbiota is closely linked to the host–disease interaction dynamics, particularly in the context of pneumonia. However, the studies included in this review employed heterogeneous methodologies, such as 16S rRNA sequencing, with notable variation in both the technological platforms used and the depth of sequencing achieved. Additionally, substantial differences in sampling sites were observed—some studies utilized bronchoalveolar lavage, while others relied on tracheal aspirates—which may have significantly influenced the detected microbial composition and diversity estimates.

A limitation of this review was the potential for publication bias and restricted access to certain relevant studies, which may have affected the generalizability of the findings. Additionally, the exclusive inclusion of literature published in English may have introduced linguistic bias. The methodological heterogeneity among the included studies further limited the ability to draw definitive conclusions. Future research should address these limitations through well-designed clinical trials and longitudinal observational studies. Moreover, additional studies are warranted to evaluate the impact of different ventilatory strategies on the lung microbiota and to explore the potential role of probiotics in modulating pulmonary dysbiosis.

### Clinical Implications and Future Directions

Evaluating the respiratory microbiota in mechanically ventilated patients is essential due to its association with reduced microbial diversity and increased risk of secondary infections. The observed dysbiosis, characterized by an imbalance between gram-positive and gram-negative organisms, contributes to adverse outcomes including ventilator-associated pneumonia and systemic inflammation.

Further research should prioritize longitudinal studies to characterize microbiota changes during and after mechanical ventilation. Identifying microbial patterns specific to intensive care settings may improve early detection of high-risk profiles. Additionally, the efficacy of microbiota-directed interventions, such as probiotics, prebiotics, and decontamination strategies, require systematic evaluation. It is also necessary to investigate how ventilatory modes, sedation protocols, and antimicrobial exposure influence respiratory microbial composition. These efforts may guide the development of targeted strategies to preserve microbial stability and reduce ventilation-associated complications.

## 5. Conclusions

Invasive mechanical ventilation is associated with alterations in the respiratory microbiome, including reduced microbial diversity and overrepresentation of genera such as *Prevotella*, *Streptococcus*, and *Pseudomonas aeruginosa*. This dysbiosis, characterized by an imbalance between gram-positive and gram-negative bacteria, can promote pneumonia, bloodstream infections, and systemic inflammatory responses, particularly in patients with pre-existing pulmonary conditions. These findings highlight the clinical impact of microbiome disruption and support the development of targeted interventions to preserve microbial homeostasis. Future research should focus on identifying predictive biomarkers and designing preventive strategies to reduce microbiome-related complications in ventilated patients.

## Figures and Tables

**Figure 1 arm-93-00023-f001:**
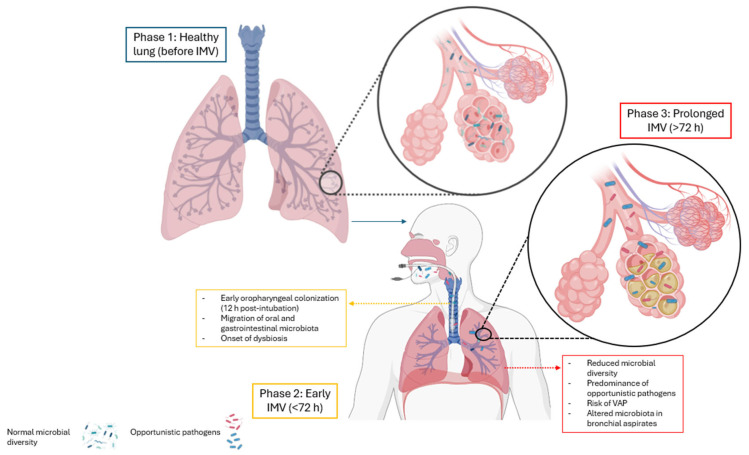
Stages of microbial dysbiosis induced by invasive mechanical ventilation. IMV: Invasive mechanical ventilation; VAP: Ventilator-associated pneumonia.

**Figure 2 arm-93-00023-f002:**
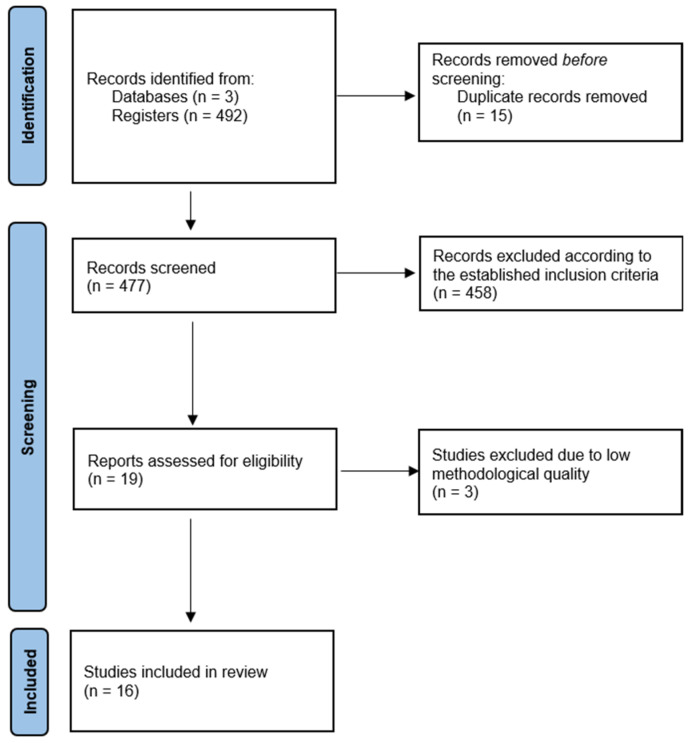
Flowchart for study selection.

**Figure 3 arm-93-00023-f003:**
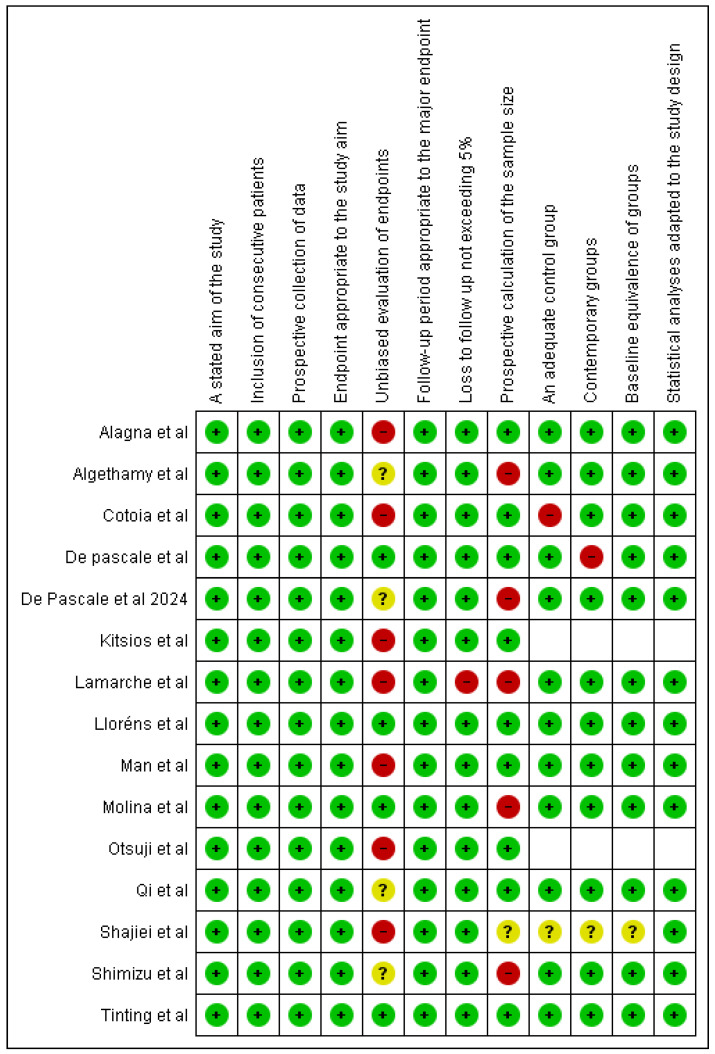
Risk-of-bias analysis. Green, yellow, and red dots indicate, respectively, low, unclear, and high risk of bias according to the MINORS scale. Colors were assigned based on the adequacy of reporting for each criterion in the included studies [18,19,20,21,22,23,24,25,27,28,29,30,31,32,33].

**Table 1 arm-93-00023-t001:** Database search strategy.

Base Data	Search Date	Search Equation	Articles Found
PubMed	3 October 2025	((((humans) AND (artificial respiration)) OR (mechanical ventilation)) AND (microbiota)) OR (dysbiosis)	183
SCOPUS	3 October 2025	((((humans) AND (artificial respiration)) OR (mechanical ventilation)) AND (microbiota))	169
Web Of Science	3 December 2025	((((humans) AND (artificial respiration)) OR (mechanical ventilation)) AND (microbiota))	140

**Table 2 arm-93-00023-t002:** Characteristics of the included studies.

Author/Year	Methodology	Population	Pathology upon ICU Admission	Mortality	Microorganisms Detected
Lamarche et al. (2018) [18]	Prospective and observational study of critically ill patients with invasive ventilatory support showing a high incidence of pneumonia in two ICUs in Hamilton, Canada.	Cohort of cases: 34 patients (14 women and 20 men). Mean age 66.6 years. Control group: 35 healthy patients.	Medical, surgical, or trauma conditions	ICU: 14.7% HOSPITAL: 35.3%.	*Enterococcus*, *Pseudomonas*, and *Staphylococcus*
Man et al. (2019) [19]	Prospective matched case–control study of hospitalized children.	Cohort of patients in the PICU: 29 patients. Cases: 154 patients (61 women and 93 men). Average age 13.6 months. Controls: 307 patients (122 women and 185 men). Average age 14.1 months.	Lower respiratory tract infections	No deaths reported.	Viruses: rhinovirus, coronavirus, respiratory syncytial virus, adenovirus. Bacteria: PICU cohort: *H. influenzae*/*H. haemolyticus*, *S. pneumoniae*, *Actinomyces* spp., and *Prevotella* spp. Controls: *Moraxella* spp, *C. propinquum*, *D. pigrum*, and *Helcococcus* spp.
Alagna et al. (2023) [20]	Exploratory analysis of data from a prospective observational study in intubated patients.	Cohort of cases: 13 patients with VAP (4 women and 9 men). Average age: 40 years. Control cohort: 22 patients with NO-VAP (9 women and 13 men). Average age: 62.5 years.	Nonpulmonary pathologies	No deaths reported.	*Streptococcus*, *Prevotella*, *Actinomyces*
Kitsios et al. (2020) [21]	Cross-sectional study from March 2015 to December 2018.	301 patients with invasive mechanical ventilation (156 men and 145 women). Average age is 59 years.	Severe respiratory insufficiency	No deaths reported.	*Streptococcus*, *Staphylococcus*, *Pseudomonadaceae*, *Stenotrophomonas*, *Prevotella*, and *Veillonella*
Outsuji et al. (2019) [22]	Prospective study conducted at the hospital of Japan University of Occupational and Environmental Health between March 2016 and March 2018.	22 patients with invasive ventilatory support (13 men and 9 women).	Aspiration pneumonia	13.6%.	*Streptococcus agalactiae*, *Enterobacter aerogene*, *Klebsiella pneumoniae*
Qi et al. (2018) [23]	Prospective study of patients hospitalized in an ICU of Ruijin Hospital, China.	Cases: 36 patients with ventilator-associated pneumonia due to *P. aeruginosa* (15 women and 21 men). Average age 68.50 years. Control: 18 healthy subjects (12 women and 6 men). The average age is 57.78 years.	Ventilator-associated pneumonia due to *P. aeruginosa*	29.6%	COHORT OF CASES: 1. Pro Group: Proteobacteria, *Pseudomonas*, *Citrobacter*, *Enterobacter*, *Klebsiella*, *Enterococcus*, and *Acinetobacter*. 2. Fir-Bac Group: Firmicutes, Bacteroidetes, Lachnospiraceae, Bacteroides, Blautia, and Alloprevotella. HEALTHY COHORT: Proteobacteria, Firmicutes, Bacteroidetes, *Prevotella*, *Streptococcus*, and *Alloprevotella*
Cotoia et al. (2023) [24]	Prospective randomized study between February 2021 and March 2022 in the Intensive Care Department of the University Hospital of Foggia, Italy.	31 patients divided into 2 groups: 15 enteral nutrition (11 men and 4 women). Average age: 56 years. 16 specialized nutrition (13 men and 3 women). Average age: 51 years.	Head trauma	At 28 days Enteral nutrition: 45%. Specialized nutrition: 25%.	*Staphylococcus* and *Acinetobacter*
De Pascale et al. (2021) [25]	Prospective observational study of patients hospitalized in two medical ICUs of Fondazione Policlínico Universitario A. Gemelli IRCCS (Rome, Italy), who developed SA-VAP between 20 March 2020, and 30 October 2020.	Total cohort: 120 patients Cases: 40 patients with COVID-19 (33 men and 7 women). Average age: 64 years Controls: 80 patients (59 men and 21 women). Average age: 62 years.	COVID-19	ICU: 25.0%. HOSPITAL: 26.7%.	*S. aureus*, *Streptococcus anginosus*, and *Olsenella*
Woo et al. (2020) [26]	Prospective study conducted in the ICU of the Sacred Heart Hospital of Chuncheon, South Korea.	Cases: 41 patients with pneumonia divided into 2 groups: Successful extubation: 22 patients (16 men and 6 women). Average age: 72 years. Failed extubation: 19 patients (12 men and 7 women). Average age: 76 years. Control group: 19 patients without pneumonia (12 men and 7 women). Average age: 76 years.	Pneumonia	Failed extubation: 63.2% Control group: 31.6%	*Pseudomonas*, *Corynebacterium*, *Streptoccocus*, *Prevotella*, and *Alloprevotella*
Shajiei et al. (2022) [27]	Pilot study between February and August 2015 of patients with mechanical ventilation was admitted to the UMCG Critical Care department.	53 patients: (32 men and 23 women). Average age: 58 years.	Neurological, respiratory, medical, cardiological, gastroenterological conditions	26.4%	*Candida*
Lloréns-Rico et al. (2021) [28]	Observational clinical trial.	58 patients whose upper respiratory tract microbiota was evaluated (13 women and 45 men). Mean age: 61 years. 35 patients whose lower respiratory tract microbiota was evaluated (12 women and 23 men). Mean age: 64 years.	COVID-19	No deaths reported.	*Staphylococcus* and *Corynebacterium*
Tingting et al. (2022) [29]	Prospective study in patients with mechanical ventilation from July 2018 to December 2019 in a tertiary hospital in adult ICUs.	52 patients: 24 with carbapenem-resistant *Acinetobacter baumannii* ventilator-associated pneumonia (CRAB-I) (16 men and 8 women). Mean age: 60.7 years. 22 with carbapenem-resistant Acinetobacter baumannii (CRAB-C) (14 men and 8 women). Mean age: 58.5 years. 6 without infection (CRAB-N) (4 men and 2 women). Mean age: 51.8 years.	Ventilator-associated pneumonia	CRAB-I: 33.3% CRAB-C: 4.5%	*Acinetobacter baumannii*
Algethamy et al. (2025) [30]	Prospective observational study in 83 ICU with invasive mechanical ventilation.	83 patients G1: VP (Pneumonia-Positive Group): 51 patients who developed VAP. G2:VN (Pneumonia-Negative Group): 32 patients who did not develop pneumonia.	Pneumonia	51 pneumonia-positive (VP) patients: 28 deaths (54.9%) 32 pneumonia-negative (VN) patients: 12 deaths (37.5%)	*Klebsiella pneumoniae* *Stenotrophomonas maltophilia* *Haemophilus influenzae* *Pseudomonas aeruginosa* *Staphylococcus aureus* *Acinetobacter baumannii*
Shimizu K et al. (2024) [31]	Retrospective, observational cohort study 76 adult ICU with invasive mechanical ventilation	Sepsis Group: 40 patients Control Group: 36 patients	Sepsis (Pulmonary origin of infection)	Sepsis Group (n = 40): Mortality: 18 patients (45%) Control Group (n = 36): Mortality: 7 patients (19.4%)	*Enterobacteriaceae* (e.g., *Escherichia coli*, *Klebsiella* spp.) *Streptococcus* spp. *Staphylococcus aureus* *Pseudomonas aeruginosa* *Acinetobacter* spp.
De Pascale et al. (2024) [32]	Prospective observational study with 70 C-ARDS patients requiring IMV	70 mechanically ventilated COVID-19 ARDS patients admitted to the ICU	C-ARDS with IVM	ICU Mortality: 48.6% (34 out of 70 patients).	*Acinetobacter*, *Staphylococcus*, *Lactobacillus*, *Klebsiella*, *Pseudomonas* *Paenibacillus*
Molina FJ et al. (2024) [33]	Retrospective Study 67 COVID-19 ICU Patient.	67 COVID-19 patients had severe pneumonia and were invasive mechanical ventilation.	COVID-19	ICU mortality 32.4% (22 patients)	*Staphylococcus aureus*, *Pseudomonas aeruginosa*, and *Acinetobacter baumannii*

ICU: Intensive care unit. PICU: Pediatric intensive care unit. VAP: Ventilator-associated pneumonia. SA-VAP: Ventilator-associated pneumonia due to *Staphylococcus aureus*.

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
