# Peer review of "Impact of Invasive Mechanical Ventilation on the Lung Microbiome"

_arm, 2025, doi:10.3390/arm93040023_

Round 1
Reviewer 1 Report
Comments and Suggestions for Authors
Overall Assessment:​​
This systematic review addresses an important and clinically relevant topic: the effects of invasive mechanical ventilation (IMV) on lung microbiota. The authors synthesize evidence from 16 studies to highlight associations between IMV, reduced microbial diversity, and increased pathogen colonization, with implications for ventilator-associated pneumonia (VAP) and mortality. While the topic is timely and the review is generally well-structured, several methodological and interpretational limitations must be addressed to strengthen the conclusions.
Major Comments:​​
​Methodological Concerns:​​
​Inclusion/Exclusion Criteria:​​
The exclusion of non-English studies introduces potential language bias. While common in systematic reviews, this should be explicitly acknowledged as a limitation.
​Quality Assessment:​​
The MINORS scale was applied, but the description of scoring is unclear.
​Heterogeneity of Studies:​​
The included studies vary widely in patient populations (e.g., pediatric vs. adult ICUs, COVID-19 vs. trauma), IMV durations, and antibiotic regimens. A discussion of how this heterogeneity impacts the synthesis of results is lacking. Consider subgroup analyses or meta-regression if feasible.
​Causality vs. Association:​​
The review emphasizes IMV as a driver of dysbiosis but does not adequately address confounding factors. For instance, antibiotic use (cephalosporins, aminoglycosides) is prevalent in ICU patients and independently alters microbiota. Statistical adjustments for antibiotics, comorbidities, or enteral nutrition are rarely described in the included studies, limiting causal inferences.
Highlight this limitation in the Discussion and caution against overinterpreting IMV as the sole contributor to dysbiosis.
​
Microbiota Analysis Techniques:​​
The methods for microbiota profiling (e.g., 16S rRNA sequencing, bronchoalveolar lavage vs. tracheal aspirates) are inconsistently reported across studies. Variations in sampling sites and sequencing depth may bias diversity estimates. Discuss how these methodological differences affect comparability.
​Clinical Implications:​​
The association between dysbiosis and mortality is noted, but the mechanisms remain speculative. Expand the Discussion to explore potential pathways (e.g., immune modulation, pathogen overgrowth) and highlight gaps in understanding causality.
The suggestion that microbiota profiling could enable early detection of dysbiosis is intriguing but unsupported by evidence from the reviewed studies. Provide specific examples or cite preliminary data if available.
​Minor Comments:​​
​Abstract:​​
​Introduction:​​
The rationale for focusing on IMV (vs. non-invasive ventilation) is unclear. Briefly justify the clinical significance of IMV-specific effects.
​Results:​​
The male predominance (64.2%) in the study population is noted but not discussed. Consider whether sex-based differences in microbiota or outcomes were explored in the included studies.
Mortality rates vary widely (e.g., 13.6% in aspiration pneumonia vs. 63.2% in failed extubation). Discuss potential reasons (e.g., differences in baseline severity).
​References:​​
Citations 11 and 12 both reference Fromentin et al. (2021), but these appear to be duplicate entries. Verify and correct.
Author Response
Dear Reviewer,
Thank you for your constructive and detailed feedback. We have addressed all your suggestions as follows:
Major Comments:
Methodological Concerns:
Inclusion/Exclusion Criteria:
The exclusion of non-English studies introduces potential language bias. While common in systematic reviews, this should be explicitly acknowledged as a limitation.
Response: This is included as a limitation within the discussion.
Quality Assessment:
The MINORS scale was applied, but the description of scoring is unclear.
Response: The elements included in the scale are collectively considered in the scoring process.
Heterogeneity of Studies:
The included studies vary widely in patient populations (e.g., pediatric vs. adult ICUs, COVID-19 vs. trauma), IMV durations, and antibiotic regimens. A discussion of how this heterogeneity impacts the synthesis of results is lacking. Consider subgroup analyses or meta-regression if feasible.
Response: The heterogeneity of the studies is discussed throughout the document and is acknowledged as a limitation. Due to the variability in the results, the findings are presented according to subtopics established by the authors, which we believe offer a preliminary framework to address the concerns raised by the reviewer and may serve as a foundation for future research on the topic.
Causality vs. Association:
The review emphasizes IMV as a driver of dysbiosis but does not adequately address confounding factors. For instance, antibiotic use (cephalosporins, aminoglycosides) is prevalent in ICU patients and independently alters microbiota. Statistical adjustments for antibiotics, comorbidities, or enteral nutrition are rarely described in the included studies, limiting causal inferences.
Highlight this limitation in the Discussion and caution against overinterpreting IMV as the sole contributor to dysbiosis.
Response: This aspect is included and addressed in the discussion.
Microbiota Analysis Techniques:
The methods for microbiota profiling (e.g., 16S rRNA sequencing, bronchoalveolar lavage vs. tracheal aspirates) are inconsistently reported across studies. Variations in sampling sites and sequencing depth may bias diversity estimates. Discuss how these methodological differences affect comparability.
Response: This aspect is acknowledged as a potential limitation, as the variability is inherent to the study designs included.
Clinical Implications:
The association between dysbiosis and mortality is noted, but the mechanisms remain speculative. Expand the Discussion to explore potential pathways (e.g., immune modulation, pathogen overgrowth) and highlight gaps in understanding causality.
The suggestion that microbiota profiling could enable early detection of dysbiosis is intriguing but unsupported by evidence from the reviewed studies. Provide specific examples or cite preliminary data if available.
Response: These aspects are addressed in the document; however, these elements are reinforced in the discussion.
Minor Comments:
Abstract:
Introduction:
The rationale for focusing on IMV (vs. non-invasive ventilation) is unclear. Briefly justify the clinical significance of IMV-specific effects.
Response: The adjustment is made in the introduction.
Results:
The male predominance (64.2%) in the study population is noted but not discussed. Consider whether sex-based differences in microbiota or outcomes were explored in the included studies.
Mortality rates vary widely (e.g., 13.6% in aspiration pneumonia vs. 63.2% in failed extubation). Discuss potential reasons (e.g., differences in baseline severity).
Response: This information is added.
References:
Citations 11 and 12 both reference Fromentin et al. (2021), but these appear to be duplicate entries. Verify and correct.
Response: The reference is different, all references are reviewed, and the respective DOI is added to each one.
We appreciate your time and consideration.
Best regards,
Authors
Reviewer 2 Report
Comments and Suggestions for Authors
The manuscript addresses a timely and clinically relevant topic—the impact of invasive mechanical ventilation on the lung microbiota. This area is of particular importance in the post-COVID-19 era, where ICU management and ventilator-associated complications remain a significant concern. The systematic review is well-structured, and the methodology is generally well-described, adhering to PRISMA guidelines. The inclusion of a PROSPERO registration is a positive aspect that adds credibility to the review process.
However, several areas require revision and clarification before the manuscript can be considered for publication: The manuscript would benefit from a thorough language and grammar check. Several sections contain awkward phrasing, inconsistencies, or unclear expressions. While the general methodology is sound, the rationale behind some inclusion/exclusion criteria (e.g., the exclusion of gray literature or why some studies scored low on the MINORS scale) should be further explained. The conclusion summarizes key findings well but would benefit from a more forward-looking perspective. You can include specific suggestions for future research directions (e.g., need for longitudinal studies, microbiota-targeted therapies, or microbiome profiling in ICU settings). I also have some additions and deletions in the text, please see and evaluate the attached file.
Best regards

The manuscript would benefit from a thorough language and grammar check. Several sections contain awkward phrasing, inconsistencies, or unclear expressions.
Author Response
Dear Reviewer,
Thank you for your constructive and detailed feedback. We have addressed all your suggestions as follows:
- A thorough language and grammar revision was carried out to ensure clarity and consistency (Full manuscript).
- We expanded the rationale behind the inclusion/exclusion criteria, particularly regarding the exclusion of gray literature and the interpretation of MINORS scores (Page 4, lines 149-155; Page 5, lines 167-173).
- The discussion section now includes forward-looking perspectives, highlighting the need for longitudinal studies, microbiota-targeted therapies, and microbiome profiling in ICU settings (Page 16-18; 4.1. Clinical implications and future directions).
- All modifications, including additions and deletions, have been implemented and are highlighted in yellow in the revised manuscript for your review.
We appreciate your time and consideration.
Best regards,
Authors

Round 2
Reviewer 2 Report
Comments and Suggestions for Authors
To authors; Thank you for corrections I suggested on your paper. Good luck with your work.